# Notch2 Regulates the Function of Bovine Follicular Granulosa Cells via the Wnt2/β-Catenin Signaling Pathway

**DOI:** 10.3390/ani14071001

**Published:** 2024-03-25

**Authors:** Wenqing Dang, Yongping Ren, Qingqing Chen, Min He, Ermias Kebreab, Dong Wang, Lihua Lyu

**Affiliations:** 1College of Animal Science, Shanxi Agricultural University, Taigu, Jinzhong 030801, China; xiaodang8660@163.com (W.D.); yongpingren797@163.com (Y.R.); chenqingq8866@163.com (Q.C.); hemin8660@163.com (M.H.); 2College of Agricultural and Environmental Sciences, University of California, Davis, CA 95616, USA; ekebreab@ucdavis.edu; 3Institute of Animal Science, Chinese Academy of Agricultural Sciences, Beijing 100193, China

**Keywords:** Notch2, Wnt2/β-catenin, bovine, follicular granulosa cells, interaction

## Abstract

**Simple Summary:**

This study investigated the crosstalk between the Wnt/β-catenin and Notch signaling pathways and the roles of these pathways in apoptosis, cell cycle progression, proliferation, and steroid hormone secretion in bovine follicular granulosa cells (GCs). Our results showed that Wnt/β-catenin and Notch pathway components regulated the development of bovine follicular GCs by modulating the expression of apoptotic, cell cycle, and steroidogenesis-related genes and proteins. Moreover, the Notch2 protein interacted with the β-catenin protein.

**Abstract:**

Ovarian follicular GCs are strongly implicated in the growth, development, and atresia of ovarian follicles. The Wnt/β-catenin and Notch signaling pathways participate in GC proliferation, differentiation, apoptosis, and steroid hormone production during follicular development. However, the crosstalk between Wnt and Notch signaling in GCs remains unclear. This study investigated this crosstalk and the roles of these pathways in apoptosis, cell cycle progression, cell proliferation, and steroid hormone secretion in bovine follicular GCs. The interaction between β-catenin and Notch2 in GCs was assessed by overexpressing *CTNNB1*, which encodes β-catenin. The results showed that inhibiting the Notch pathway by Notch2 silencing in GCs arrested the cell cycle, promoted apoptosis, reduced progesterone (P4) production, and inhibited the Wnt2-mediated Wnt/β-catenin pathway in GCs. IWR-1 inhibited Wnt2/β-catenin and Notch signaling, reduced GC proliferation, stimulated apoptosis, induced G1 cell cycle arrest, and reduced P4 production. *CTNNB1* overexpression had the opposite effect and increased 17β-estradiol (E2) production and Notch2 protein expression. Co-immunoprecipitation assays revealed that Notch2 interacted with β-catenin. These results elucidate the crosstalk between the Wnt/β-catenin and Notch pathways and the role of these pathways in bovine follicular GC development.

## 1. Introduction

The ovarian follicle is formed by granulosa cells (GCs) surrounding the oocyte. Follicular development involves the proliferation and differentiation of GCs, and morphological and functional alterations in these cells determine the follicular developmental stages. During this process, GCs produce nutrients, metabolites, and molecular signals that regulate the quiescence, activation, and death of oocytes [1,2] and the production of steroid hormones in the ovary [3]. Hence, GC proliferation and differentiation lead to follicular growth, which is indispensable for oocyte development and female fertility. Follicular atresia is associated with DNA fragmentation and GC apoptosis. After ovulation, GCs differentiate to form the corpus luteum, which secretes progesterone and maintains pregnancy. The primary regulators of GC function are cytokines and hormones, including gonadotropins and insulin-like growth factor, from multiple signaling pathways [4].

Signaling pathways are conserved across species, and Wnt and Notch signaling governs cell proliferation [5], differentiation [6], and apoptosis [7]. When active, the Wnt/β-catenin pathway contains extracellular Wnt ligands, frizzled receptors, low-density lipoprotein receptors, and β-catenin, which is an effector protein [8]. β-catenin is translocated to the nucleus, where it combines with transcription factors of the T-cell factor/lymphoid enhancer binding factor family to promote the transcription of downstream target genes [9,10,11]. Notch-mediated juxtacrine signaling occurs when Notch ligands (Jagged1/2 and Delta-like1/3/4) bind to Notch 1/2/3/4 receptors, leading to the cleavage of Notch intracellular domains (NICDs) by γ-secretase. NICD is translocated to the nucleus and interacts with transcription factors RBP-J and Mastermind co-activator to activate transcription [12,13]. Components of these two pathways are expressed in GCs and are involved in follicular development. Further, the abnormal expression of some of these components impairs follicular development and reduces female fertility [11,12,14,15,16], potentially leading to ovarian cancer [17,18].

There is crosstalk between Wnt/β-catenin, Notch, and other signaling pathways [19]. Loss-of-function assays found a synergistic effect between Wnt1 and Notch in *Drosophila* wingless mutants, demonstrating that the Wnt/β-catenin and Notch pathways interact with each other [20,21]. These pathways affect hair follicle maintenance [22], rhombic pattern formation [23], and somitogenesis in vertebrates [24]. In addition, these pathways have compensatory roles in maintaining the homeostasis of rat luteal cells [25] and are implicated in the proliferation and differentiation of intestinal and embryonic stem cells [26,27].

These two pathways act synergistically or antagonistically, depending on the cellular environment and connections with other pathways [28]. However, the interactions between these two pathways in GCs remains unclear. Follicle-stimulating hormone (FSH) regulates β-catenin protein and *Wnt2* mRNA expression in bovine GCs, suggesting the involvement of Wnt/β-catenin signaling in follicular development and ovarian steroidogenesis in cattle [3]. The Notch2 receptor and Jagged2 ligand are expressed in bovine GCs and thecal cells, participating in luteinized GC development [29]. These results imply that Wnt/β-catenin and Notch signaling are synergistically regulated in bovine follicular GCs.

In this study, we assessed: (1) the inhibitory effect of the Wnt/β-catenin or Notch signaling pathways on the key factors of another pathway, and whether there is a synergistic effect between these two pathways on GC proliferation and apoptosis, as well as the expression and secretion of steroid hormones, and (2) the interactions between Notch2 and β-catenin pathway proteins.

## 2. Materials and Methods

### 2.1. Ethical Approval

All animal procedures were conducted in accordance with the guidelines of the China Council on Animal Care. This animal study was reviewed and approved by the Ethics Committee of Shanxi Agricultural University (Approval No. SXAU-EAW-2021C.WR.0040100150).

### 2.2. Isolation and Culture of GCs

Thirty-five ovaries from healthy cows were collected from an abattoir in Wenshui (Shanxi, China) and preserved in Dulbecco’s phosphate-buffered saline (DPBS) supplemented with 1% (*v*/*v*) penicillin–streptomycin liquid (100 IU/mL–0.1 mg/mL, Gibco, Shanghai, China). Then, the ovaries were washed thrice with DPBS, and the tissues attached to the ovaries were removed. The follicles (5–8 mm) were dissected from the ovary, and GCs were isolated, as described previously [30]. The GCs were seeded in 96-well or 6-well plates containing DMEM-F12 medium (Hyclone, Shanghai, China), 10% fetal bovine serum (Cellmax, Beijing, China) and 1% penicillin–streptomycin liquid, and were incubated in a humidified incubator with 5% CO_2_ at 37 °C.

### 2.3. Treatment of GCs

The effect of IWR-1 on GC proliferation, cell cycle progression, and apoptosis was evaluated. The GCs were seeded in 96-well plates (5.0 × 10^3^ cells/well) for cell proliferation or 6-well plates (1.0 × 10^5^ cells/well) for the other assays until 60% confluence was reached. The supernatant was removed, and the cells were cultured in DMEM-F12 complete medium containing IWR-1 (MCE, Shanghai, China) (0, 1, 2.5, 5, 10, or 20 μM). Cell viability was measured using the Cell Counting kit-8 (CCK-8). Cell cycle progression and apoptosis in GCs treated with 5 μM IWR-1 were analyzed using flow cytometry. Gene and protein expression were quantified using real-time quantitative polymerase chain reaction (RT-qPCR) and Western blotting techniques. The effect of IWR-1 on Wnt and Notch expression, steroidogenesis gene expression, and steroid hormone production was also evaluated. The cells were centrifuged at 3000 rpm for 15 min at 4 °C, and the concentrations of progesterone (P4) and 17β-estradiol (E2) were measured in the supernatant.

The effect of treatment with a lentiviral vector expressing an siRNA targeting Notch2 was evaluated. Notch2 siRNA was synthesized by GengPharma (Shanghai, China) using the following primers: sense, 5′-GATCCGCTATGAGCCTTGTGTAAATGTTCAAGAGACATTTACACAAGGCTCATAGCTTTTTT-3′ and antisense, 5′-AATTCAAAAAAGCTATGAGCCTTGTGTAAATGTCTCTTGAACATTTACACAAGGCTCATAGCG-3′. A negative control (NC) was synthesized using the following primers: sense, 5′-GATCCGTTCTCCGAACGTGTCACGTTTCAAGAGAACGTGACACGTTCGGAGAACTTTTTTG-3′ and antisense, 5′-AATTCAAAAAAGTTCTCCGAACGTGTCACGTTCTCTTGAAACGTGACACGTTCGGAGAACG-3. Lentiviral packaging and centrifugation were performed as described previously [29]. The GCs were cultured in 6-well plates until 60% confluence was reached and were transfected in medium containing a final concentration of 10 μg/mL of polybrene. Transfection efficiency was assessed at 48 h post-transfection by measuring green fluorescence intensity. The GCs transfected with siRNA targeting Notch2 were selected with 3μg/mL puromycin and were harvested for RNA and protein extraction. P4 and E2 concentrations were measured in cell supernatants.

The interactions between β-catenin and Notch2 pathway proteins were analyzed. To overexpress *CTNNB1*, the coding sequence was amplified using PCR and cloned into the *Xba*I and *Bam*HI sites of the pHBLV-CMV-MCS-EF1-ZsGreen-T2A-puro lentiviral vector (Hanbio, Shanghai, China). Overexpression vector packaging and transduction were performed as described above. The GCs were cultured for 48 h and harvested for protein extraction. The interactions between β-catenin and NICD2 proteins were analyzed using co-immunoprecipitation assays. Apoptosis protein expression and steroid hormone secretion were also examined.

### 2.4. CCK-8 Assays

The GCs were treated with different concentrations of IWR-1 for 48 h, as described above. Cell viability was measured using the CCK-8 kit (Dojindo, Shanghai, China) according to the manufacturer’s instructions. Briefly, 100 μL of medium containing 10% CCK-8 reagent was added to each well and incubated for 4 h. Absorbance was measured at 450 nm using a microplate reader. The cell proliferation rate was calculated using the formula [OD (treated) − OD (blank)]/[OD (control) − OD (blank)] × 100%.

### 2.5. Cell Cycle and Apoptosis Assays

The GCs were cultured in 6-well plates in medium containing 0 or 5 μM IWR-1 for 48 h and were digested with 0.25% trypsin (Gibco, Shanghai, China). Cell cycle and apoptosis assays were performed using the Cell Cycle Assay Kit (Yeasen Biotechnology, Shanghai, China) and the AnnexinV-FITC/PI apoptosis Assay kit (Zeta Life, Shanghai, China), as described previously [31]. Cell cycle distribution and apoptosis were analyzed using a flow cytometer (NovoCyte, ACEA Biosciences, San Diego, CA, USA) and data were analyzed using NovoExpress version 1.5.0.

### 2.6. Western Blotting

The GCs were lysed in RIPA buffer (Beyotime, Shanghai, China) containing 1% (*v*/*v*) PMSF (Boster, Wuhan, China) for 30 min on ice. The lysates were centrifuged at 13,000 rpm for 10 min at 4 °C. Protein concentrations were quantified using a BCA protein assay kit (Boster, Wuhan, China). The proteins (10 µg) were separated using 8% or 10% SDS-PAGE and transferred to polyvinylidene difluoride membranes (Boster, Wuhan, China). The membranes were blocked with 5% non-fat dry milk in Tris-buffered saline (TBS) for 1 h at room temperature. Then, the membranes were incubated with rabbit polyclonal primary antibodies against NICD2 (1:1000, Cell Signaling Technology, Boston, MA, USA), β-catenin (1:1000, Cell Signaling Technology, Boston, MA, USA), Bax (1:500, Bioss, Beijing, China), and caspase-3 (1:500, Bioss, Beijing, China) overnight at 4 °C. After washing with TBS, the membranes were incubated with IRDye^®^ 800CW goat anti-rabbit IgG secondary antibody (1:18,000, LI-COR Biosciences, Shanghai, China) for 1 h at room temperature. The blots were imaged using an Odyssey laser imaging system (LI-COR Biosciences, Lincoln, NE, USA) and quantified using ImageJ software version 1.8.0. The protein levels were normalized to β-actin (1:10,000, BioWorld, Nanjing, China).

### 2.7. RT-qPCR

Total RNA was extracted from the GCs using RNAiso Plus (Takara, Dalian, China) and was reverse transcribed using the PrimeScript RT Reagent Kit with gDNA Eraser (Takara, Dalian, China). RT-qPCR was performed in a 20-μL reaction volume containing TB Green^®^ Premix Ex Taq^™^ II (Tli RNaseH Plus) (Takara, Dalian, China) using the CFX96 system (Bio-Rad Laboratories, Inc., Hercules, CA, USA). The amplification conditions consisted of an initial denaturation step at 95 °C for 60 s, followed by 40 cycles at 95 °C for 30 s, 95 °C for 5 s, and Tm for 30 s. The transcription level of each target gene was normalized to RPLP0 using the 2^−ΔΔCt^ method. The primers used in PCR amplification are listed in Table 1.

### 2.8. Measurement of E2 and P4 Concentrations by ELISA

The concentrations of E2 and P4 in the GC supernatants were assayed using ELISA kits (Blue Gene, Shanghai, China), following the manufacturer’s instructions. The lowest detectable concentration of E2 and P4 was 1.0 pg/mL and 0.1 ng/m. The samples were diluted as necessary. Steroid hormone concentrations were normalized to 100,000 cells and were calculated using a standard curve.

### 2.9. Co-Immunoprecipitation (co-IP) Assays

The GCs were transduced with a *CTNNB1* overexpressing lentivirus and stably transfected cells were selected using 3 μg/mL puromycin. Total protein was extracted as described above and was quantified using a BCA protein assay kit (Thermo Fisher Scientific, Waltham, MA, USA). Co-immunoprecipitation was performed using a co-immunoprecipitation kit (Thermo Fisher Scientific, Waltham, MA, USA) with rabbit polyclonal antibodies against β-catenin (1:50, Cell Signaling Technology, Boston, MA, USA), Notch2 (1:200, Cell Signaling Technology, Boston, MA, USA), and IgG (1:50, Abclonal, Wuhan, China). β-catenin and Notch2 protein expression was measured using Western blotting.

### 2.10. Statistical Analysis

GC proliferation was analyzed using one-way analysis of variance followed by Tukey’s multiple comparison tests, and pairwise comparisons were performed using *t*-tests in SPSS version 24.0 (IBM, New York, NY, USA). Cell proliferation assays were performed in sextuplicate and were repeated independently three times. A *p*-value of less than 0.05 was considered statistically significant.

## 3. Results

### 3.1. Effects of IWR-1 on GC Function and Wnt/β-Catenin and Notch Pathways

IWR-1 decreased GC proliferation based on absorbance at 450 nm, and the decrease was more pronounced starting at 2.5 μM (Figure 1A). The Western blots showed that 5 μM IWR-1 strongly decreased β-catenin expression in a dose-independent manner (Figure 1B, Appendix A). Therefore, this concentration was used in subsequent experiments. Moreover, 5 μM IWR-1 arrested the cell cycle and downregulated the cell cycle genes in GCs. Specifically, IWR-1 increased the proportion of GCs in the G1 phase and decreased the percentage of GCs in the G2 phase, indicating that IWR-1 induced G1 cell cycle arrest (Figure 1C). The RT-qPCR analysis showed that IWR-1 downregulated *CDK4* and *CCND2* and upregulated *p21* (Figure 1D), which was consistent with the flow cytometry results. IWR-1 promoted apoptosis by increasing the percentage of early apoptotic cells by 8.25% (Figure 1E, quadrant 2-2). Further, IWR-1 increased the mRNA and protein expression of Bax and Caspase-3 and decreased the protein expression of *Bcl-2* and the *Bcl-2*/*Bax* ratio (Figure 1F, Appendix A).

IWR-1 modulated the Wnt/β-catenin and Notch pathways in GCs. IWR-1 decreased the mRNA and protein expression of Notch2 (Figure 2, Appendix A) and decreased the protein expression of β-catenin (Figure 1B), but not the mRNA expression of its gene, *CTNNB1* (Figure 2B). In the Wnt pathway, IWR-1 downregulated *Wnt2* (a ligand gene) and *LEF1* (a target gene). In contrast, IWR-1 upregulated *Axin2* (Figure 2B). In the Notch pathway, IWR-1 downregulated *Hey2* (a target gene) and upregulated *Jag1* (a Notch ligand) (Figure 2B).

The effect of IWR-1 on the expression of steroidogenesis genes and hormone secretion by GCs was assessed by measuring the mRNA expression of steroidogenesis genes and E2 and P4 concentrations. IWR-1 decreased the concentration of P4, but not E2, in cell supernatants (Figure 3A,B) by downregulating *CYP11A1* and *3β-HSD* and upregulating *STAR* (Figure 3C). The intra-assay coefficient of variation (CV) for E2 and P4 was 4.09% and 8.70%, respectively, in the supernatant of GCs not treated with IWR-1, and 8.81% and 7.94%, respectively, in the supernatant of GCs transduced with 5 μM IWR-1.

### 3.2. Effects of Notch2 Silencing on GC Function and Wnt/β-Catenin and Notch Pathways

The GCs were transduced for 48 h with a lentivirus carrying Notch2 siRNA or NC siRNA and were imaged using fluorescence microscopy. The Notch2 siRNA and NC groups exhibited strong green fluorescence, indicating successful plasmid integration into the GCs (Figure 4).

*Notch2* silencing increased the apoptosis rate in GCs. Specifically, *Notch2* silencing increased the protein (Figure 5A,B, Appendix A) and mRNA (Figure 5C) expression of Bax and Caspase-3 and decreased the expression of the anti-apoptotic gene *Bcl-2* (Figure 5C). Furthermore, *Notch2* silencing arrested the cell cycle by downregulating *CCND1*, *CCND2*, and *CDK4* and upregulating *p21* (Figure 5D).

The Western blots showed that Notch2 siRNA significantly decreased NICD2 and β-catenin protein expression (Figure 6A, Appendix A). The RT-qPCR analysis showed that Notch2 siRNA downregulated *Wnt2*, *CTNNB1*, *LEF1* (a Wnt target gene), *Notch2*, and *Hey2* (a Notch target gene) (Figure 6B).

The ELISAs showed that Notch2 siRNA decreased the P4 concentrations in GCs (Figure 7B,C). The RT-qPCR analysis showed that Notch2 siRNA downregulated *STAR*, *CYP11A1*, and *3β-HSD*, which was consistent with the steroid hormone concentration results (Figure 7A). The intra-assay CV for E2 and P4 was 8.29% and 8.08%, respectively, in the supernatant of GCs transduced with lenti-NC, and 7.59% and 4.24%, respectively, in the supernatant of GCs transduced with lenti-Notch2.

### 3.3. Interaction between β-Catenin and Notch2 in GCs

The GCs were transduced for 48 h with a lentivirus overexpressing *CTNNB1* (lenti-OE-*CTNNB1*) or NC (lenti-OE-NC). The lenti-OE-CTNNB1 and lenti-OE-NC groups exhibited strong green fluorescence, indicating successful plasmid integration into the GCs (Figure 8).

*CTNNB1* overexpression decreased the expression of the apoptotic protein Caspase-3 and Bax in the GCs (Figure 9, Appendix A), increased E2 concentration, and decreased P4 concentration (Figure 10). The intra-assay CV for E2 and P4 was 8.69% and 3.60%, respectively, in the supernatant of GCs transduced with lenti-OE-NC, and 7.76% and 2.81%, respectively, in the supernatant of GCs transduced with lenti-OE-CTNNB1.

*CTNNB1* overexpression increased the protein expression of β-catenin and NICD2 (Figure 11A,B, Appendix A). To determine the interaction between β-catenin and NICD2 proteins in GCs, β-catenin was used as bait to obtain immunoprecipitated protein complexes to which these proteins could bind. The complexes were identified using Western blotting. β-catenin and NICD2 were expressed in the cells transduced with lenti-OE-CTNNB1 or lenti-OE-NC, but not in the cells in which IgG antibody was used as a control, suggesting that β-catenin and NICD2 interact with each other (Figure 11C, Appendix A).

## 4. Discussion

Wnt and Notch pathway components are expressed in the postnatal ovary and affect follicular development by regulating the proliferation of GCs [11,32,33,34].

IWR-1 inhibits the Wnt/β-catenin pathway by stabilizing the Axin destruction complex, resulting in the degradation of β-catenin [35]. We found that 5 μM IWR-1 reduced GC proliferation, which is consistent with the finding that 5 μM IWR-1 decreased HCT116 colorectal cancer cell proliferation in vitro [36]. However, 0.5, 1, and 10 μM IWR-1 did not significantly decrease the number of bovine GCs [30,37]. A possible explanation for this discrepancy is that previous studies cultured GCs in serum-free medium, whereas we cultured GCs in medium containing 10% fetal bovine serum. It was shown that IWR-1 decreased the proliferation of human SO-RB50 cells, induced G1 arrest by decreasing the mRNA and protein expression of CCND1, and increased the rate of apoptosis [38]. As IWR-1 concentration increased, the percentage of GCs in the G1 phase increased, *CDK4* and *CCND2* expression decreased, and *p21* expression increased, indicating that the Wnt/β-catenin pathway inhibited GC proliferation by regulating the cell cycle. Apoptosis, as a failsafe measure during the cell cycle, ensures the fidelity and quality of cell proliferation. Flow cytometry results revealed that 5 μM IWR-1 increased the apoptosis rate of GCs by increasing the mRNA and protein expression of Bax and Caspase-3 and by decreasing the mRNA expression of *Bcl-2*. These results suggest that IWR-1 induced mitochondrial apoptosis in GCs by inhibiting Wnt/β-catenin signaling. Conversely, *CTNNB1* overexpression inhibited apoptosis by decreasing Caspase-3 and Bax protein expression, suggesting that apoptosis in GCs is activated by the canonical Wnt pathway.

The Wnt/β-catenin pathway regulates ovarian follicle maturation and steroid hormone production [39]. For instance, the intravitreal injection of IWR-1 into dominant and subordinate follicles in vivo altered steroid hormone production and decreased the estrogen-to-progesterone ratio in ovarian follicles [40]. In contrast, the Wnt/β-catenin pathway stimulated E2 synthesis in GCs in medium and large buffalo follicles, whereas IWR-1 decreased E2 levels in GCs from medium follicles [37]. In addition, IWR-1 decreased the FSH-induced production of E2 in bovine GCs in vitro [40]. We found that the in vitro treatment of GCs with 5 μM IWR-1 decreased the concentration of P4, but not E2, and downregulated *3β-HSD* and *CYP11A1*, which are involved in P4 synthesis. A possible explanation is that FSH induction is required for E2 secretion by GCs, as FSH regulates the activity of aromatase (encoded by *CYP19A1*), which converts androgens to estrogen. *CTNNB1* overexpression increased E2 levels and decreased P4 concentrations, which is consistent with the finding that *CTNNB1* is upregulated in bovine large antral follicular GCs with higher E2 concentrations than in follicles with lower E2 concentrations [3]. Moreover, *CTNNB1* deletion in mouse primary GCs inhibited *CYP19A1* expression and E2 production, demonstrating that *CYP19A1* is a target of *CTNNB1* in GCs [41].

We found that 5 μM IWR-1 upregulated *Axin2* and decreased β-catenin protein expression and *wnt2* and *LEF1* gene expression. This result suggests that 5 μM IWR-1 inhibits the Wnt/β-catenin pathway in GCs. Studies have shown that the Wnt2 protein is implicated in follicular development and GC proliferation. The RNAi-mediated knockdown of *Wnt2* inhibits GC proliferation [42], whereas *Wnt2* overexpression reverses this effect [43]. Wnt2 regulates mouse GC proliferation through β-catenin [44], and FSH controls the expression of both proteins in bovine GCs [3]. In addition, consistent with our results, IWR-1 prevented FSH-induced GC proliferation, reduced E2 secretion, increased Axin2 protein levels, and decreased β-catenin expression [30], demonstrating that Axins inhibit this pathway, and inhibiting this pathway upregulated Axin genes.

The Notch pathway is implicated in follicle histogenesis and GC proliferation and survival, and the treatment of isolated ovaries with the inhibitors DAPT and L-685,458 reduces GC proliferation [45], demonstrating that GC proliferation depends on Notch signaling [42]. Notch2 is expressed in GCs and oocytes during follicular development, suggesting that Notch2, as the primary Notch receptor, is involved in oocyte–GC communication and the recognition of GCs. Notch2 increases GC proliferation in follicles cultured in vitro. Conversely, *Notch2* knockdown increases the number of atretic ovarian follicles, whereas the simultaneous knockdown of *Jag1* and *Notch2* decreases GC proliferation [12]. Notch2 controls GC proliferation by regulating c-Myc expression [12]. *Notch2* silencing downregulated the cell cycle genes *CCND1*, *CCND2*, and *CDK4* and upregulated *p21*, which is consistent with previous findings in bovine luteinized GCs [29]. Furthermore, the Notch2-mediated inhibition of the Notch pathway caused apoptosis in GCs by upregulating *Bax*, *Caspase-3*, and *p21* and downregulating *Bcl-2*, similar to the way that the Wnt signaling pathway regulates GC apoptosis. Therefore, Notch signaling promotes follicular GC development by controlling cell proliferation, cell cycle progression, and apoptosis.

Notch signaling also affects steroid hormone production [46,47]. *Notch2* expression in primary mouse GCs cultured in vitro peaked at day 5 of culture, whereas GCs increased the secretion of E2 and P4 by upregulating *STAR*, *CYP19A1*, and *CYP11A1* [46]. DAPT treatment decreased the number of GCs and E2 secretion in sheep [48]. Our results suggest that the Notch pathway, mediated by Notch2, increases P4 secretion by upregulating *STAR*, *3β-HSD,* and *CYP11A1*. However, there were no detectable changes in E2 concentrations and the relative expression of *CYP19A1*, which regulates E2 synthesis. It has been shown that the rate of proliferation of primary mouse GCs decreased with culture time, leading to the appearance of differentiated preovulatory GCs by inhibiting intercellular contacts [46]. Therefore, we speculate that the Notch2-mediated suppression of Notch signaling induces the differentiation of GCs into luteinized GCs, consequently decreasing P4 concentration, which is consistent with the changes in GC morphology in the later stages of culture. DAPT decreased P4 concentrations in murine luteinized cells by downregulating *CYP11A1*. NICD3 overexpression increased P4 secretion in human chorionic gonadotropin-induced GCs by upregulating *CYP11A1* and *STAR* [45], whereas P4 production decreased in gonadotropin-responsive mouse GCs transfected with NICD1 or NICD2 [49]. The downregulation of *STAR*, *3β-HSD*, and *CYP11A1*, which regulate P4 synthesis, is consistent with the above results.

We found that *Notch2* silencing downregulated Notch2 protein and its target gene *Hey2*, indicating that the Notch pathway was successfully inhibited. The *hey2* gene is a member of the basic helix-loop-helix family, which is a transcriptional repressor and a target of Notch signaling. DAPT downregulated *Hey2* in fetal and neonatal mouse ovaries [50]. In addition, the shRNA-mediated knockdown of *Notch2* in bovine luteinized GCs decreased the mRNA and protein expression of Notch2 and the relative expression of *Hes1* and *Hey2* [29].

There is crosstalk between the Notch and Wnt/β-catenin pathways in several cell types during animal development [19]. This interaction may be synergistic or antagonistic, depending on the environmental stimuli [51]. Moreover, the mechanisms of interaction between proteins from these pathways are classified into three categories: the co-regulation of transcriptional targets, the influence of the transcriptional targets of a pathway on those of another pathway, and direct interaction between pathways [52]. The latter mechanism has been studied the most. We found that IWR-1 decreased the expression of Notch2 protein and the target gene *Hey2* and upregulated *Jag1*. However, the reason for this phenomenon is unclear. *CTNNB1* overexpression increased β-catenin and NICD2 protein expression. In turn, *Notch2* silencing downregulated *Wnt2*, *CTNNB1*, and *LEF1*. The consistent expression of key genes across pathways suggests a synergistic effect between these pathways in bovine GCs cultured in vitro. β-catenin inhibition by Notch promotes β-catenin degradation in lysosomes upon Notch dephosphorylation without the need for γ-secretase release from NICDs or the expression of target genes [53]. NICDs inhibit β-catenin activity by preventing β-catenin from binding to its target site, ultimately inhibiting the Wnt/β-catenin pathway [54]. Furthermore, *Jag1* knockdown enhances the proliferation of mouse preovulatory GCs, and represses GC differentiation by decreasing the expression of enzymes and factors involved in steroid synthesis and secretion [46], which is consistent with our results.

To investigate the molecular interactions between the two pathways in vitro, the Wnt/β-catenin pathway was activated by overexpressing *CTNNB1*, and the interaction between β-catenin and Notch2 was assessed using co-immunoprecipitation. The results showed that NICD2 was present in the protein complexes obtained using a β-catenin antibody as bait, suggesting an interaction between these two proteins. However, these interactions in GCs are unclear. The protein complex composed of RBP-J, NICD, and β-catenin induces the generation of arterial endothelial cells by vascular progenitor cells [55]. β-catenin proteins with reduced transcriptional activity promote ovarian cancer cell proliferation by upregulating *Jagged1* in the Notch signaling pathway [28]. Notch interacted with β-catenin and inhibited its activity by targeting the complex to the proteasome. Nonetheless, the interactions between catenin and NICD, and their roles in GCs, warrant further investigation. Furthermore, GSK3β, a member of the destruction complex, mediates the interaction between Wnt and Notch signaling and directly modulates Notch signaling to promote carcinogenesis in the colon and neural progenitor cells [56]. GSK3β also activates NICD2 phosphorylation and downregulates its target genes [52,57]. *Hes1* expression is regulated by β-catenin [58]. Research is underway to elucidate the interactions between these two pathways.

## 5. Conclusions

This study demonstrated that inhibiting the Wnt/β-catenin or Notch signaling pathways in bovine follicular GCs in vitro regulates follicular GC function by modulating cell cycle progression and apoptosis, and by reducing P4 secretion. Moreover, these two pathways may act synergistically through β-catenin and Notch2 protein interactions. These results elucidate the roles of these pathways in bovine follicular GC development.

## Figures and Tables

**Figure 1 animals-14-01001-f001:**
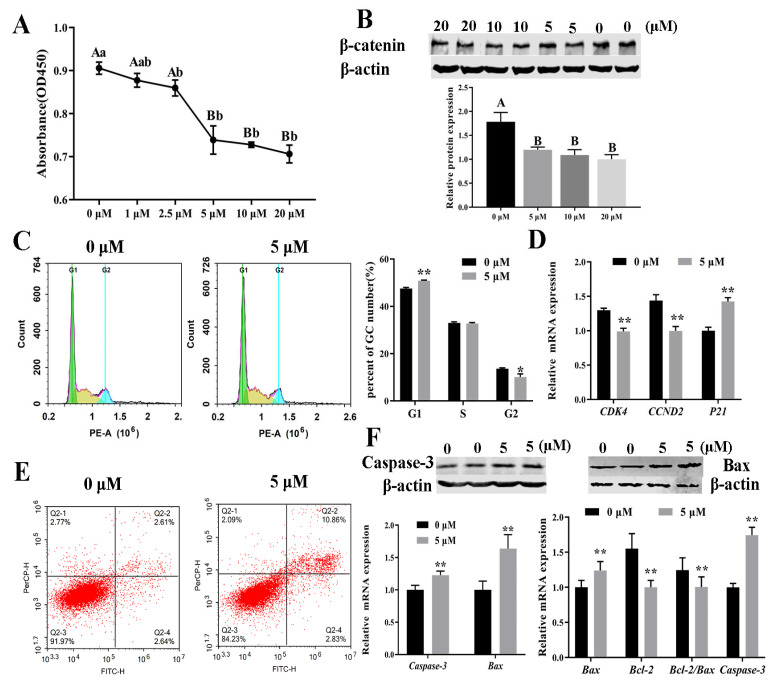
Effect of IWR-1 on the proliferation, cell cycle progression, and apoptosis of bovine follicular granulosa cells (GCs). (**A**) Effect of IWR-1 on cell proliferation; (**B**) effect of IWR-1 on β-catenin protein expression; (**C**) effect of IWR-1 on the percentage of cells at the G1, S, and G2 phases; (**D**) effect of IWR-1 on the expression of cell cycle genes (*CDK4*, *CCND2*, and *P21*) in GCs; (**E**) flow cytometric analysis of the effect of IWR-1 on the percentage of apoptotic cells (quadrants 2-2 and 2-4); (**F**) effect of IWR-1 on the expression of cell apoptosis genes (*Bax*, *Bcl-2*, and *Caspase-3*) and apoptosis proteins (Bax and Capase-3); * *p* < 0.05, ** *p* < 0.01. Different lowercase and uppercase letters indicate significant differences at *p* < 0.05 or *p* < 0.01, respectively.

**Figure 2 animals-14-01001-f002:**
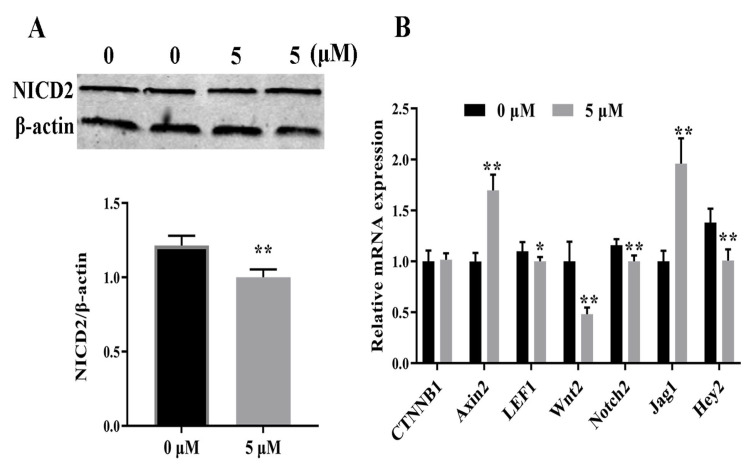
Effect of IWR-1 on the Wnt/β-catenin and Notch pathways in bovine follicular granulosa cells (GCs). (**A**) Effect of IWR-1 on NICD2 protein expression; (**B**) effect of IWR-1 on the expression of Wnt genes (*CTNNB1*, *Axin2*, *LEF1*, and *Wnt2*) and Notch genes (*Notch2*, *Jag1*, and *Hey2*). * *p* < 0.05, ** *p* < 0.01.

**Figure 3 animals-14-01001-f003:**
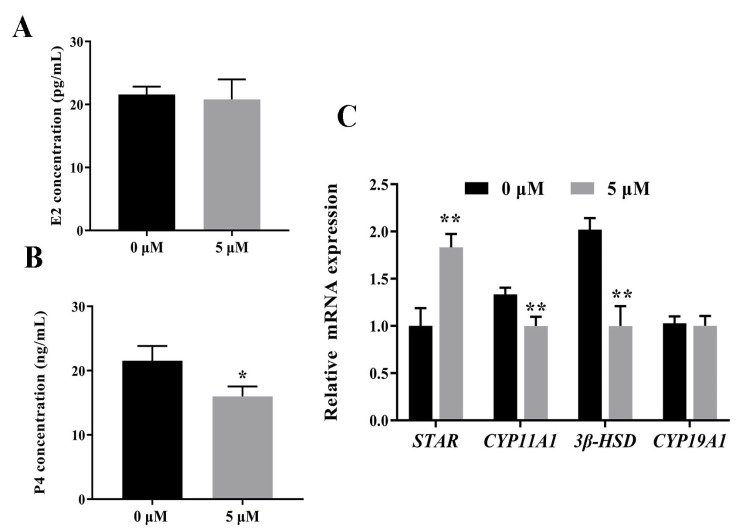
Effect of IWR-1 on steroid hormone concentrations in the supernatant of bovine follicular granulosa cells (GCs) and the expression of steroidogenesis genes in GCs. (**A**,**B**) 17β-estrogen (E2) and progesterone (P4) concentrations in the cell supernatant; (**C**) relative mRNA expression of *STAR*, *CYP11A1*, *3β-HSD*, and *CYP19A1* in GCs. * *p* < 0.05, ** *p* < 0.01.

**Figure 4 animals-14-01001-f004:**
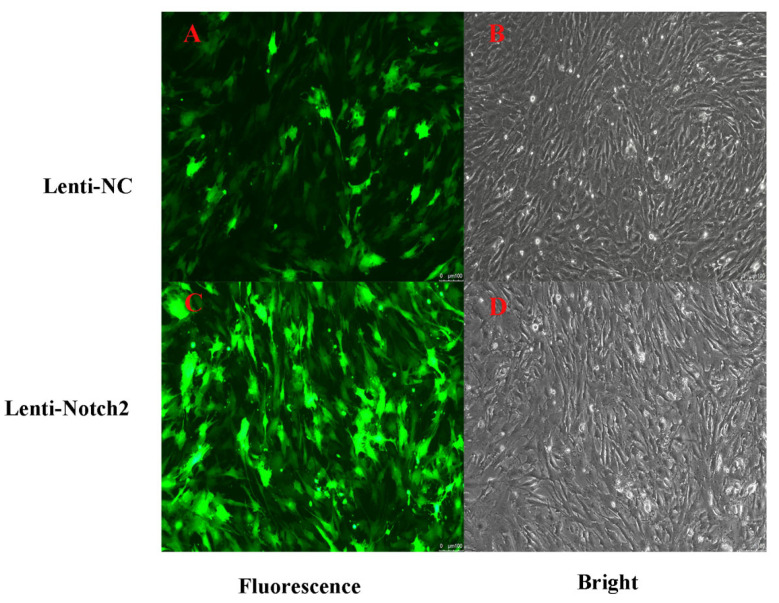
Bright-field and fluorescence images of bovine follicular granulosa cells (GCs). GCs were transfected with negative control siRNA (**A**,**B**) or *Notch2* siRNA (**C**,**D**) (×40).

**Figure 5 animals-14-01001-f005:**
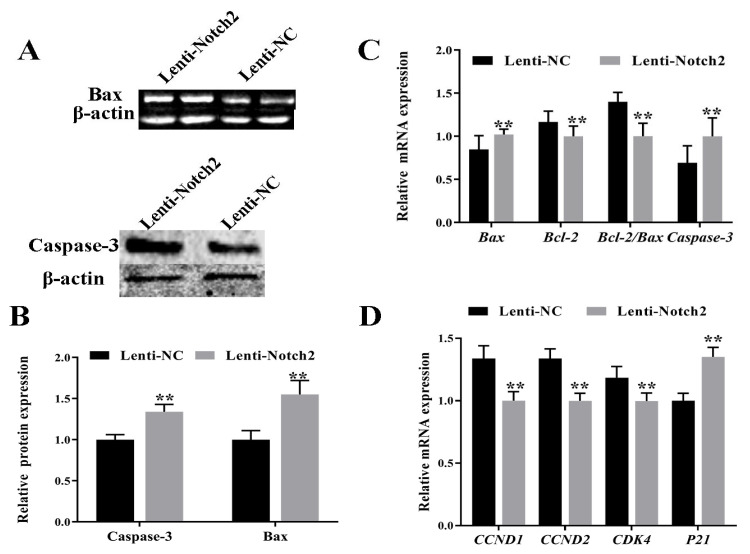
Effect of *Notch2* silencing on the expression of apoptosis and cell cycle genes in bovine follicular granulosa cells (GCs). (**A**,**B**) Expression of apoptosis proteins (Bax and Caspase-3); (**C**,**D**) expression of apoptosis genes (*Bax*, *Bcl-2*, and *Caspase-3*) and cell cycle genes (*CCND1*, *CCND2*, *CDK4*, and *P21*). ** *p* < 0.01.

**Figure 6 animals-14-01001-f006:**
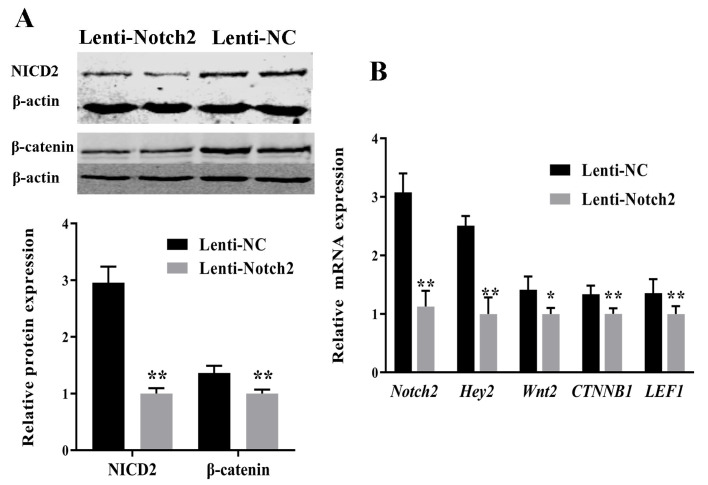
Effect of *Notch2* silencing on the expression of Wnt and Notch pathway genes and proteins in bovine follicular granulosa cells (GCs). (**A**) Effect of *Notch2* silencing on NICD2 and β-catenin protein expression; (**B**) effect of *Notch2* silencing on the expression of Wnt genes (*CTNNB1*, *LEF1*, and *Wnt2*) and Notch genes (*Notch2* and *Hey2*). * *p* < 0.05, ** *p* < 0.01.

**Figure 7 animals-14-01001-f007:**
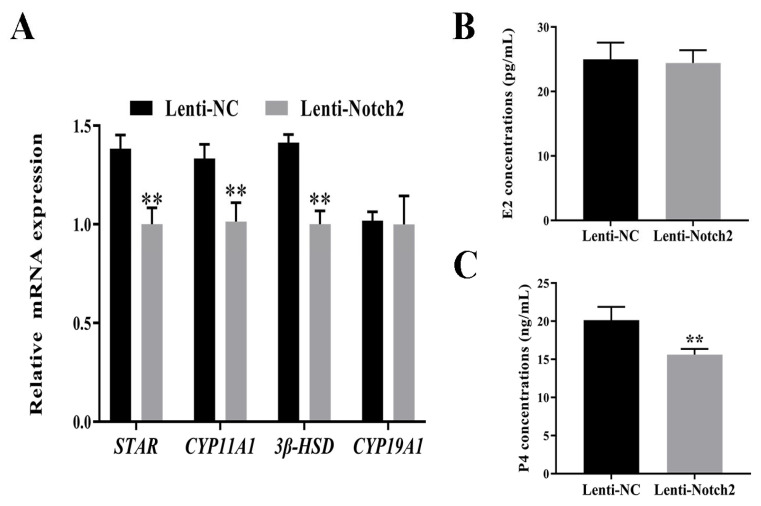
Effect of *Notch2* silencing on steroid hormone concentrations in the supernatant of bovine follicular granulosa cells (GCs) and the expression of steroidogenesis genes in GCs. (**A**) Expression of genes involved in steroid hormone synthesis (*STAR*, *CYP11A1*, *3β-HSD*, and *CYP19A1*); (**B**,**C**) 17β-estrogen (E2) and progesterone (P4) concentrations in the supernatant. ** *p* < 0.01.

**Figure 8 animals-14-01001-f008:**
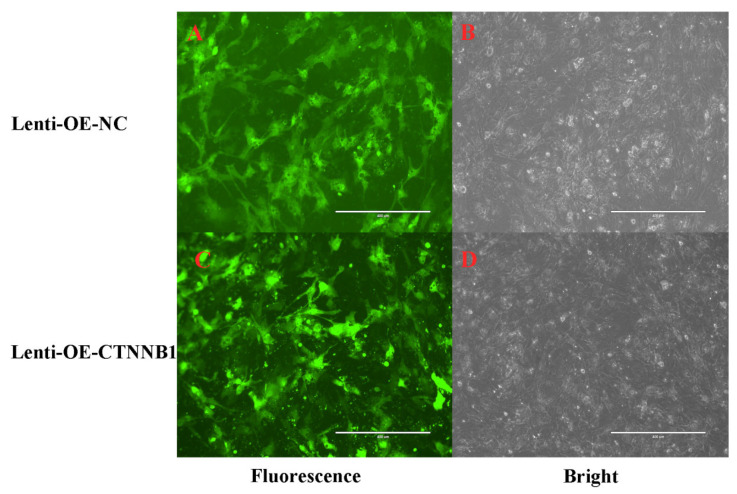
Bright-field and fluorescence images of bovine follicular granulosa cells (GCs) transduced with a negative control lentivirus (**A**,**B**) or a lentivirus overexpressing *CTNNB1* (**C**,**D**) (100×).

**Figure 9 animals-14-01001-f009:**
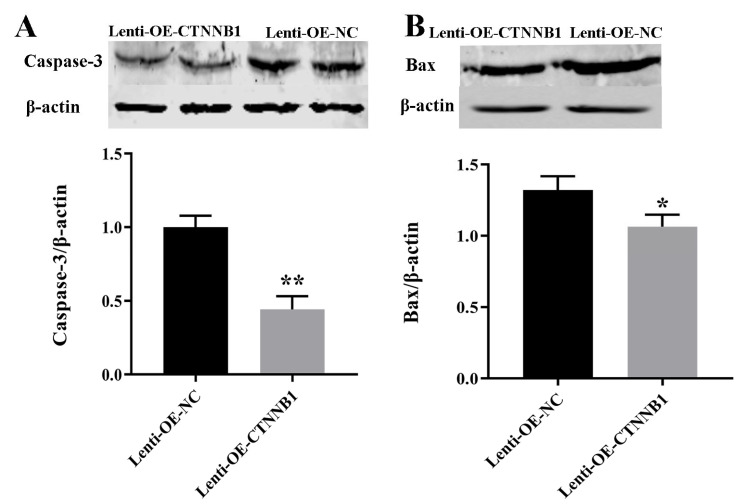
Effect of *CTNNB1* overexpression on the expression of apoptotic proteins in bovine follicular granulosa cells (GCs). (**A**) Caspase-3 expression; (**B**) Bax expression. * *p* < 0.05, ** *p* < 0.01.

**Figure 10 animals-14-01001-f010:**
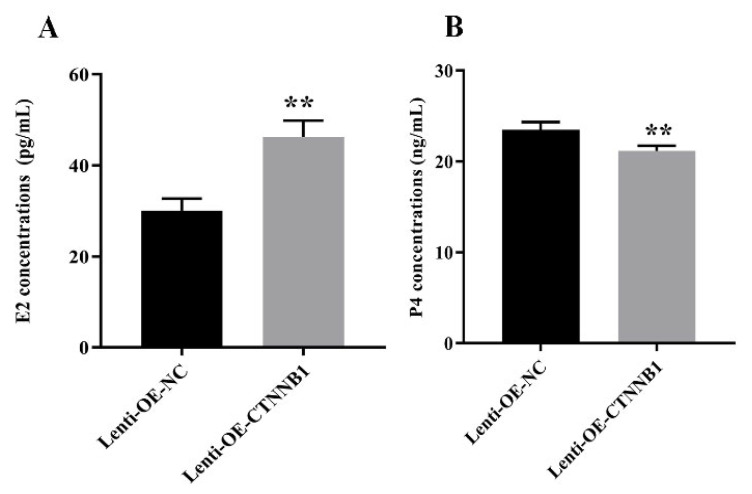
Effect of *CTNNB1* overexpression on steroid hormone concentrations in the supernatant of bovine follicular granulosa cells (GCs). (**A**,**B**) Concentrations of 17β-estrogen (E2) and progesterone (P4) in the cell supernatant. ** *p* < 0.01.

**Figure 11 animals-14-01001-f011:**
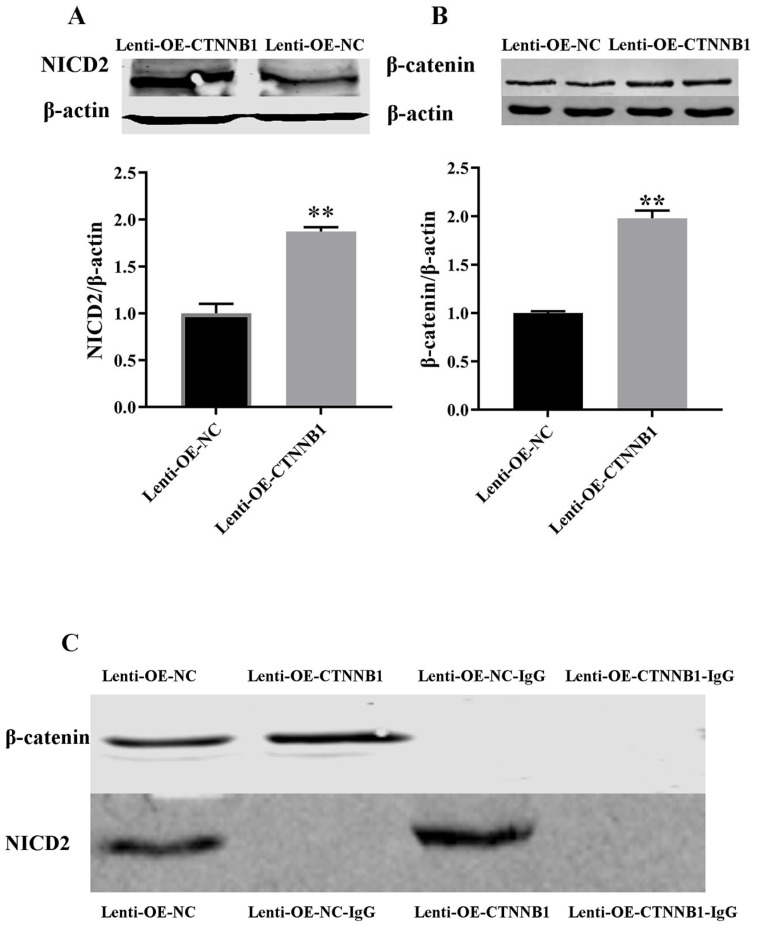
Interaction between β-catenin and Notch2 in bovine follicular granulosa cells (GCs). (**A**,**B**) Effect of *CTNNB1* overexpression on NICD2 and β-catenin protein expression in GCs; (**C**) protein expression of β-catenin and NICD2 in protein complexes. ** *p* < 0.01.

**Table 1 animals-14-01001-t001:** Primers used in RT-qPCR.

Gene	GenBank Accession Number	Primer Sequence	Product Size (bp)
*Wnt2*	NM_001013001.1	F:5′-GAACCGCCAAGGATAACAAG-3′	88
		R:5′-ACAAACGCTCTGGCAAACTT-3′	
*CTNNB1*	NM_001076141.1	F:5′-AGATGATGGTGTGCCAAGTG-3′	109
		R:5′-AGATGACGAAGGGCACAGAT-3′	
*LEF1*	NM_001192856.1	F:5′-GCGAATGTCGTAGCTGAGTG-3′	121
		R:5′-CCTTCCGCGCTAATTCATAA-3′	
*AXIN2*	NM_001192299	F:5′-AGCGGATACAGGTCCTTCAG-3′	111
		R:5′-GTCACTGGATATCTCGCTGTC -3′	
*Notch2*	NM_002686114.6	F:5′-AGACGGCCTAACACCAAGAG-3′	80
		R:5′-CTGTTCCCCTTGGCATCCTT-3′	
*Hey2*	NM_001192055.1	F:5′-TCTGAGTTGAGACGACTGGTG-3′	143
		R:5′-GCGTGTGCATCAAAGTAGCC-3′	
*Jag1*	NM_001191178.1	F:5′-GGTCAATGGCGAGTCCTTCA-3′	76
		R:5′-GTCGTTGGTGTTCTGTGTGC-3′	
*Caspase-3*	NM_001077840.1	F:5′-AGCCATGGTGAAGAAGGAATC-3′	89
		R:5′-CTGCAATAGTCCCCTCTGAAG-3′	
*Bax*	NM_173894.1	F:5′-GACATTGGACTTCCTTCGAGA-3′	126
		R:5′-AGCACTCCAGCCACAAAGAT-3′	
*BCL-2*	NM_001166486.1	F:5′-GTGGATGACCGAGTACCTGAAC-3′	124
		R:5′-AGACAGCCAGGAGAAATCAAAC-3′	
*CDK4*	NM_001037594.2	F:5′-CCTTCATGCCAACTGCATCG-3′	148
		R:5′-CCAGAGTGTAACAACCACAGGT-3′	
*CCND1*	NM_001046273.2	F:5′-GCACGACTTCATCGAGCACT-3′	115
		R:5′-GAACTTCACGTCTGTGGCA-3′	
*CCND2*	NM_001076372.1	F:5′-CACCGATGTGGATTGCCTCA-3′	117
		R:5′-TCCAGCTCATCCTCCGACTT-3′	
*P21*	NM_001098958.2	F:5′-CGGTGGAACTTCGACTTTGT-3′	183
		R: 5′-CAAGTGGTCCTCCTGAGACG-3′	
*CYP11A1*	NM_176644.2	F: 5′-CACCGATATTATCAGAAACCC-3′	249
		R: 5′-ATTGGTGATGGACTCAAAGG-3′	
*CYP19A1*	NM_174305.1	F: 5′-CACCCATCTTTGCCAGGTAGTC-3′	78
		R:5′ACCCACAGGAGGTAAGCCTATAAA-3′	
*HSD3β*	NM_174343.3	F: 5′-TGCCACAATCTGACCGCATC-3′	167
		R:5′- CTCCACCAACAGGCAGATGA-3′	
*STAR*	NM_174189.3	F:5′-CAGAAGGGTGTCATCAGAGCG-3′	169
		R:5′-CAAAATCCACCTGGGTCTGC-3′	
*RPLP0*	NM_001012682.1	F:5′-CAACCCTGAAGTGCTTGACAT-3′	227
		R:5′-AGGCAGATGGATCAGCCA-3′	

## Data Availability

The raw data supporting the conclusions of this article will be made available by the authors upon request.

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
