# Peer review of "Notch2 Regulates the Function of Bovine Follicular Granulosa Cells via the Wnt2/β-Catenin Signaling Pathway"

_animals, 2024, doi:10.3390/ani14071001_

Round 1

Reviewer 1 Report

Comments and Suggestions for Authors

The authors goal was to better understand the role of Notch signaling and the relevant signalling pathway in bovine granulosa cells.  The work is hard to assess as the figures and figure labels are at times too small to read.  In one figure (figure 3) the authors have reversed the findings in the text from what is presented in the figure.  They then spend significant space describing a result that is different than presented.  English utilization can be significantly improved and I will not go through all of the examples here as they are too numerous.  Improved presentation of the results would be required to fully evaluate the work.

Comments on the Quality of English Language

Throughout there are many situations where English utilization is poor.  In some cases it is hard to discern what the authors are trying to convey such that suggestions to improve utilization is difficult.

Reviewer 2 Report

Comments and Suggestions for Authors

This manuscript describes the results of an assay linking the Notch and Wnt/catenin pathways in bovine granulosa cells in vitro. Many techniques are performed to analyze steroidogenesis, proliferation, apoptosis and cell differentiation, These techniques allow you to achieve the work objectives . Although the work is interesting, and the results are original, it requires a rewriting, since it becomes very difficult to understand, especially in the discussion.

The introduction includes the more recent bibliography related with the work, but in the last paragraph the author must present clearly the objective of the research without introduce the techniques that will be used to reach it.
Description of material and methods is correct and complete
In results, the figures are very small and, consequently, is very difficult their interpretation

The discussion is poor. The authors do not relate the different results to each other and, furthermore, they do not hypothesize how these results could be transferred to knowledge about the live animal
In the discussion (lines 321-324) the authors write that” Flow cytometry results revealed that 5 μM IWR-1 increased the apoptosis rate of GCs by downregulating the expression of Bax and caspase-3 and Bcl-2. in the GCs”. It seems contradictory that apoptosis increases if the expression of Bax and Caspase-3 decreases.

Comments on the Quality of English Language

English must be reviewed throughout the manuscript. Even the sections that do not need to be modified due to their content such as materials and methods, and results, are difficult to understand due to the language.

Reviewer 3 Report

Comments and Suggestions for Authors

The MS entitled “Notch2 Regulates the Function of Bovine Follicular Granulosa Cells via Wnt2/β-catenin Signaling Pathway” by Wenqing Dang et al investigate the interaction of Notch2 and β-catenin pathways on the apoptosis, cell cycle, proliferation and steroid hormone secretion in bovine follicular GCs. The authors got a wealth of data to support this argument by using inhibitor IWR-1 on Wnt pathway and lentiviral vector expressing small interfering RNA (siRNA) targeting Notch and overexpressing CTNNB1, RT-qPCR, western blot and co-immunoprecipitation assays. However, there are still some deficiencies in the paper, as follows:

General comments

Some of the results are not well consistent with the figure, and the writhing also needs to be a little improved.

Specific comments

L30: Need a conclusion in the abstract and clarify the significance of this study.

L32: Key words: misspelled  “granulose”

L44: redundant “such as”

L85: Incomplete bracket

L171: for the ELISA, how about the cv% (Coefficient of variation), pls supply this data.

L209: Figure 1B, are there any significant differences among 5,10, 20 μM IWR-1? need further comparison and identification.

L233: decreased the concentration of P4, but not E2, is Not consistent with the figure 3AB. In addition, Figure 3 A B the hormone level normalized to the cell number? not the volume (corresponding to L175). upregulated STAR? Is also Not consistent with the figure 3C, pls check.

L261: downregulated Wnt2, is Not consistent with the figure 6B, pls check

L269: better to arrange the steroidogenic proteins based on the order of hormone synthesis as STAR, CYP11A1, 3β-HSD (for P4 synthesis) and CYP19A1 (for further E2 synthesis) in the figure and the text.

Comments on the Quality of English Language

Moderate editing of English language required.

Round 2

Reviewer 1 Report

Comments and Suggestions for Authors

The authors seek to understand the some of the potential roles of the Wnt2/beta-catenin and Notch2 signaling in bovine granulosa cells.  Similar to other mammalian systems (rodents in particular) the signalling pathways are involved in granulosa cell function and inhibition or over expression of components change the cell cycle progression, proliferation and steroidogenesis by these cells.

The authors have improved and corrected the figures and and improved the English usage.  There is still room for improvement in English usage and I give a couple of examples below.

The work does a thorough job of elucidating impacts of altering the signalling cascade on a variety of cellular functions at RNA and protein levels.  The findings are in line with what would be expected based on what we know about the signalling cascade in other systems.

Specific comments

L25  cell cycle progression

L55 pathway is contains

L113 cell cycle progression

L137 Lentiviruses packaging to Lentiviral packaging...

L156 In short, Briefly, medium...

Comments on the Quality of English Language

English usage has been improved, but could still be improved further.

Reviewer 2 Report

Comments and Suggestions for Authors

In the current form the manuscript can be accepted
